# Modeling Human Cytomegalovirus in Humanized Mice for Vaccine Testing

**DOI:** 10.3390/vaccines8010089

**Published:** 2020-02-17

**Authors:** Johannes Koenig, Sebastian J. Theobald, Renata Stripecke

**Affiliations:** 1Laboratory of Regenerative Immune Therapies Applied, Excellence Cluster REBIRTH, Hannover Medical School, 30625 Hannover, Germany; Koenig.Johannes@mh-hannover.de (J.K.); theobald.sebastian@mh-hannover.de (S.J.T.); 2Clinic of Hematology, Hemostasis, Oncology and Stem Cell Transplantation, Hannover Medical School, 30625 Hannover, Germany; 3German Center for Infection Research (DZIF), Partner Site Hannover-Braunschweig, 30625 Hannover, Germany

**Keywords:** CMV, HHV-5, humanized mice, HIS, transplantation, T cells, antibodies, dendritic cells, vaccines

## Abstract

Human cytomegalovirus (HCMV or HHV-5) is a globally spread pathogen with strictly human tropism that establishes a life-long persistence. After primary infection, high levels of long-term T and B cell responses are elicited, but the virus is not cleared. HCMV persists mainly in hematopoietic reservoirs, whereby occasional viral reactivation and spread are well controlled in immunocompetent hosts. However, when the immune system cannot control viral infections or reactivations, such as with newborns, patients with immune deficiencies, or immune-compromised patients after transplantations, the lytic outbursts can be severely debilitating or lethal. The development of vaccines for immunization of immune-compromised hosts has been challenging. Several vaccine candidates did not reach the potency expected in clinical trials and were not approved. Before anti-HCMV vaccines can be tested pre-clinically in immune-compromised hosts, reliable in vivo models recapitulating HCMV infection might accelerate their clinical translation. Therefore, immune-deficient mouse strains implanted with human cells and tissues and developing a human immune system (HIS) are being explored to test anti-HCMV vaccines. HIS-mice resemble immune-compromised hosts as they are equipped with antiviral human T and B cells, but the immune reactivity is overall low. Several groups have independently shown that HCMV infections and reactivations can be mirrored in HIS mice. However, these models and the analyses employed varied widely. The path forward is to improve human immune reconstitution and standardize the analyses of adaptive responses so that HIS models can be forthrightly used for testing novel generations of anti-HCMV vaccines in the preclinical pipeline.

## 1. Introduction

Human cytomegalovirus (HCMV or HHV5) is a human-specific latent pathogen and belongs to the family of beta-herpesviruses. The estimated frequency of HCMV-seropositive individuals in the global general population is 83% (95% uncertainty interval: 78%–88%), and in the subpopulation of blood or organ donors, it is 86% (95% uncertainty interval: 82%–89%) [1]. Congenital or neonatal HCMV infection during pregnancy can frequently cause mental retardation or hearing loss in newborns [2]. Although innocuous to most people during adult life, HCMV has been associated with several pathologies and chronic conditions that worsen with age [3,4,5]. The latent HCMV reservoir can be found in CD14^+^ monocytes, CD34^+^ stem cells, and some epithelial cells from bone marrow and salivary glands [6,7], whereas epithelial cells, including endothelial cells, are lytic targets [3]. The latent HCMV infection phase is characterized by minimal gene expression and none or very low production of new infective viral particles. Once HCMV is triggered to enter the lytic stage, a fast and complex interaction of different viral genes and proteins orchestrate a boost in viral production while dampening down the host’s innate and adaptive immune responses [7,8,9]. The lytic phase is divided in the immediate-early (IE), early (E), and late (L) phases. Along these phases, the episomal viral genome is actively replicated, which results in massive cell death. Eventually, some of the cells infected are spared from cell death and establish viral latency, characterized by the silencing of IE1 and IE2 gene transcription factors [10]. In humans, HCMV lytic reactivation can be clinically diagnosed by an increase of the viral DNA copies detectable in blood. This can be associated, in extreme cases, with severe inflammation and tissue damage of the liver, lung or colon [11]. The pathways leading to HCMV reactivation are not completely understood; however, several cytokines seem implicated, such as granulocyte macrophage colony-stimulating factor (GM-CSF), granulocyte colony-stimulating factor (G-CSF), interleukine-4 (IL-4), tumor-necrosis-factor-alpha (TNF-α), and interferon-alpha (IFN-α) [6].

Most latently infected immune-competent individuals contain antigen-specific memory T cells and produce antiviral neutralizing antibody responses against HCMV. Yet, for immune-compromised individuals undergoing hematopoietic stem cell transplantation (HCT) or solid organ transplantation (SOT), HCMV can reactivate out of the latent reservoir and induce severe problems like graft failure, hepatitis, pneumonia, colitis, retinitis or esophagitis [11,12]. The incidence of HCMV reactivation in the context of HCT affects more than 50% of seropositive recipients (R^pos^) and 10% of seronegative recipients (R^neg^) [13]. If a donor is HCMV seronegative (D^neg^), this is an important clinical criterion for being rejected as a potential allogeneic HCT donor. This is due to the fact that in the absence of memory immune responses, there are high chances of HCMV reactivation in the frail recipient. The current pharmacologic standard of care to treat reactivation is the use of antiviral drugs (such as ganciclovir, foscarnet, and cidofovir), but the side-effect profile is broad [11,14]. Clinical trials testing the new generation drug letermovir showed strong anti-viral potency and no myelosuppression, but the efficacy of this drug in the high-risk R^pos^/D^neg^ HCT group lacking proper memory immune control still remains to be established [15,16].

The development of efficacious anti-HCMV vaccines could potentially provide immune control against HCMV to lower the reactivation risks in transplanted patients or deleterious effects on newborns. As further discussed below, several vaccine candidates have been tested in clinical studies, such as attenuated viruses, DNA vaccines, recombinant proteins, and vectored vaccines [17,18,19]. Yet, after several completed trials, no vaccine has reached the desired potency. This review describes different modalities of humanized mouse models of HCMV infections and how human vaccines can eventually be tested for potency prior to clinical use.

## 2. Mouse Models of HCMV Infections and Human Immune Responses 

The strict tropism of HCMV to humans urged the study of homologous cytomegaloviruses (CMVs) in other animal species. There is vast literature regarding murine CMV (MCMV) and also several reports about rhesus monkey CMV (RhCMV) models (for a brief overview, see Table 1*)* [20,21,22,23,24,25,26,27,28]. This wealth of work contributes extensively to the understanding of basic aspects of the interactions between CMV and the host. However, preclinical evaluation of human vaccines in these models has so far been limited due to evolutionary divergences of the immune system in different species and the closely associated functional co-evolution of CMV proteins [29,30]. Potency testing of anti-HCMV vaccines in vitro through short-term assays was used as an alternative [31,32,33,34,35]. These strategies can be informative regarding the ability of antigens provided in different vaccine formulations to re-stimulate memory T cells in vitro. However, these rapid potency assays are not yet able to mimic de novo primary stimulation of naïve T and B cells since the spatio-temporal immunological requirements, such as cell migration and localization in the microenvironment of lymphatic tissues, is missing.

During the last thirty years, the emergence and progress of mouse models developing the human immune system (HIS) have incrementally contributed to the immunological understanding of human infections and cancer [45]. HIS mouse models basically require two main components: (i) mouse hosts that allow efficient engraftment and persistence of the xenografted human cells, and (ii) human precursors or differentiated immune cells that can adequately engraft, bio-distribute and self-maintain after administration. HCMV in vivo models were contemporarily improved, depending on the severely immune-deficient mouse strains available. Initially, HIS models explored mice derived from the CB17 strain carrying a spontaneous mutation in the protein kinase-DNA activated-catalytic polypeptide (Prkdc^scid^), which were called “Severe comcined immunodeficiency (SCID)-mice” [46]. These mice could be engrafted short-term with hematopoietic stem cells (HSCs), peripheral blood mononuclear cells (PBMCs), and fetal tissues, but the persistency of murine Natural killer (NK) cells and residual T cell receptor (TCR) rearrangements in murine T cells and BCR in B cells resulted in rejections of human tissues. SCID mice backcrossed with the non-obese diabetic (NOD) mouse strain encoding the mutated SIRPα protein resulted in the “NOD-SCID” strain, with reduced phagocytosis of human cells [47,48]. A further improvement became possible when the mouse IL-2 receptor gamma was knocked out, fully abolishing the development of mouse NK, T, and B cells and resulting in the Nod.SCID.Gamma (“NSG”) strain [49]. An equivalent of the NSG strain is the Nod.Rag.Gamma (“NRG”) strain, which instead of the spontaneous *scid* mutation, harbors a targeted knock-out of the recombinant of activating gene 1 (RAG1), abolishing the maturation and rearrangement T and B cell receptors [50,51]. In order to further optimize the engraftment of human cells, these immune-deficient mice are commonly irradiated or treated with myeloablating drugs. Regarding the source of human immune cells, human peripheral blood (huPBL) or selected mature T cells have been commonly used, but the caveat is that the T cells will invariantly cause xenograft graft-versus-host disease (xeno-GVHD), leading to mouse lethality 2 to 4 weeks later. An alternative is to transplant mice with selected human CD34^+^ HSCs, resulting in the development of human T cells trained endogenously in the mouse thymus, and rarely leading to xeno-GVHD. Cord blood (CB) is a practical source of CD34^+^ HSCs since this human material can be obtained frequently and upon demand and the CD34^+^ can be conveniently cryopreserved after isolation without major loss of engraftment. CD34^+^ HSCs can also be obtained from the bone marrow or leukapheresis from adult donors, but procurement is more complex and the frequency of CD34^+^ HSCs is lower.

Theobald et al. recently demonstrated the straightforward feasibility of HCMV modeling in vivo using NRG mice with long-term reconstitution of the human immune system (NRG-HIS) (Figure 1a, Table 1). CB-CD34^+^ HCT into female NRG mice resulted in consistent and persistent (>25 weeks) development of functional and mature T and B cell responses with rare incidences of xeno-GVHD. Seventeen weeks after HCT, NRG-HIS mice were administered i.p. with MRC-5 fibroblasts infected with a traceable HCMV laboratory strain secreting *Gaussia* luciferase (GLuc) [38,52]. Non-invasive optical imaging analyses were performed to distinguish HCMV infection and HCMV reactivation (promoted by daily s.c. administration of recombinant human G-CSF for seven days). HCMV reactivations after G-CSF administrations were quantitatively measured by optical imaging analyses and quantification of viral genome copies in different tissues (such as spleen, lymph nodes, liver, salivary glands, bone marrow). In parallel, the development and tissue bio-distribution of human T and B cells were followed by flow cytometry analyses. Interestingly, the immune phenotypic data demonstrated a noticeable discrepancy between HCMV infections and reactivations. Remarkably, T cell development was enhanced in thymus upon HCMV infection, which was associated with the expansion of memory CD4^+^ and CD8^+^ T cells in secondary lymphatic tissues and up-regulation of the programmed cell death (PD)-1 activation marker on T cells. Additionally, upon HCMV infection and reactivation, increased levels of follicular T helper cells (T_FH_) were detectable in HIS-mice. This was correlated with significantly higher levels of class-switched IgG^+^ B cells in spleen, liver and bone marrow. Further, using a more sophisticated bioinformatics approach, principal component analyses (PCA) and linear discriminant analyses (LDA) of the organ datasets enabled multivariate statistical analyses. HCMV infection and reactivation could be identified immunologically in the model by clustering specific sets of immune phenotypic markers. T cells isolated from NRG-HIS mice after HCMV infection or reactivation showed functional recall responses in vitro against several HCMV antigens, such as IE1, pp65, and gB. HCMV-specific IgM and IgG humoral responses were detected in plasma (Table 1). In conclusion, the phenotypic and functional immune parameters obtained from this model, aided technologically by dynamic optical imaging and multicolor for cytometry and bioinformatics data quantitation, makes it a very good platform to test new vaccines.

Smith et al. described the use of normal human dermal fibroblasts (NHDF) infected with the HCMV TRpM1A strain. The cells were injected i.p. into NSG mice, and then the mice were implanted with osmotic pumps for constant hG-CSF release [37]. The numbers of HCMV genome viral copies in peripheral blood, spleen, liver, and kidney were increased after HCMV reactivations (Table 1) [37].

Similar to these models, HCT was performed using CD34^+^ HSCs isolated from G-CSF-mobilized peripheral blood obtained from HCMV-seropositive donors. Using this approach, Hakki et al. confirmed the transmission of the HCMV infection in spleen and liver of the recipient mice [39].

Tomic et al. used CD34^+^ stem cells isolated from fetal liver to transplant NSG mice expressing the HLA*A02 molecule (NSG-A2). The HCMV-TB40 wild type strain or an attenuated strain expressing a recombinant ULBP2 protein for activation of the NKG2D receptor were used [40]. The infection of humanized NSG-A2 mice was performed by i.p. injection of in vitro infected monocyte-derived dendritic cells (DCs). G-CSF was used to promote viral reactivation. The mice infected with the attenuated mutant strain expressing ULBP2 showed higher NK cell numbers compared to mice infected with wild type TB40, and this was associated with decreased HCMV viral copies. Further, both strains promoted virus-specific CD8^+^ T cell and IgM responses (Table 1) [40].

Besides the use of CD34^+^ cells on their own, the combined transplantation of fetal liver-derived CD34^+^ HSCs, fetal liver, and fetal thymus tissues was explored to generate the bone marrow–liver and thymus (hBLT) model. “BLT mice” can show high levels of human immune reconstitution, but sporadic early onset of xeno-GVHD can be observed [41,53]. Besides, a general major obstacle of BLT models is the ethical perception regarding the procurement of fetal tissues for the establishment of xenografted mice.

Mocarski et al. were pioneers in the use of fetal tissues to study HCMV in vivo and used human fetal thymus and liver transplants into CB17 *scid*/*scid* mice [43]. The implanted human tissues were inoculated with the HCMV Toledo virus strain and infection was detected up to 35 days after virus inoculation. Additionally, they confirmed that viral replication occurred in epithelial cells, which could be suppressed by treatment with ganciclovir [43].

Crawford et al. (Figure 1b, Table 1) described the use of the clinical HCMV TR strain or the TB40/GFP laboratory strain. Similarly to Smith et al., in vitro infected NHDF cells were administered i.p. into NSG-BLT mice and reactivation was stimulated with osmotic pump release of hG-CSF. Besides the HCMV infection profile, they additionally demonstrated human CD4^+^ and CD8^+^ T cell responses against the viral IE1 and pp65 proteins [42]. They were also able to detect human antibody responses in the plasma with in vitro HCMV neutralization capacity [42].

A model additionally exploring lung tissue was described by Wahl et al. (Figure 1c, Table 1). This model, called NSG-BLT-L, was obtained after implantation of human lung tissue into NSG-BLT mice [44]. The rationale for this approach was to explore epithelial, endothelial, and mesenchymal cells, which might serve as lytic tissue targets for HCMV replication. The HCMV strains, TB40-Luciferase, AD169, and ADrUL131, were inoculated directly into the engrafted lung tissue. The HCMV immediate-early, early, and late proteins were detectable by immunohistochemistry. Pre-treatment of mice with ganciclovir abolished HCMV infection of the transplanted lung. Next, they compared the HCMV infection in the lung implants for NSG-L versus NSG-BLT-L mice. HCMV infections could be confirmed for both models, and since decreased luciferase signals were detectable in NSG-BLT-L mice, this suggested that this model conferred better immunological control of the viral spread. In order to confirm this, HCMV-specific CD8+ T cell response against IE1 and pp65 viral antigens was shown in the NSG-BLT-L. After repeated inoculation of HCMV into NSG-BLT-L mice, mimicking chronic antigen exposure, they detected HCMV-specific IgG responses with potent in vitro neutralization capacity [44].

When deciding on one type of these engraftment models for HCMV research or for vaccine testing, the obvious question arises, whether the more complex “multi-organ”-humanized mice (e.g., NSG-BLT-L), or simpler models (such as the NRG-HIS) should be favored. Until now, this question could not be answered since immune responses or vaccinations have not been directly compared.

In addition to the variations of the tissue engraftment approaches, the mouse host may also cause an impact. Several immune-deficient mouse strains have been more recently created, showing improved developments of the innate and adaptive immune system. A triple transgenic mouse, with the knock-ins of human transgenes expressing IL-3/GM-CSF/stem-cell factor (SCF), has been generated based on the NSG background. Upon CD34^+^ transplantations, these mice showed increased levels of myeloid cells and dendritic cells in the circulation. Furthermore, these mice showed an increased number of regulatory T cells (T_reg_) [54]. Similarly, the “MISTRG-6” strain developed for expression of human M-CSF, IL-3, IL-6, GM-CSF, and thrombopoietin (TPO) also sowed improved T, B, and NK cell development after humanizations [55,56]. Besides the transgenic expression of human cytokines, another area of intensive development is to improve the human adaptive immune response in the NSG and NRG strains by providing the expression of human HLA class I and II molecules, and matching the stem cell donor [57,58]. Thus, these new strains should also be considered for the development of HCMV infection models.

Incidentally, another approach to stimulate the human immune reconstitution in humanized mice is to provide additional human-derived dendritic cells (DCs) matched to the CD34^+^ stem cell donor. Since the generation of dendritic cells is laborious, a recently developed approach is a monocyte-based engineered cell therapy to accelerate the human T and B cell reconstitution for antiviral protection. After a short lentiviral transduction protocol (24 h), the CD14^+^ monocytes obtained from adult or cord blood express GM-CSF, INF-α and antigens [59]. This genetic manipulation induced the monocytes to self-differentiate into highly viable (>2 weeks) DCs [59]. Administration of these induced DCs expressing the pp65 antigen into humanized mice resulted in improved immune regeneration with enhanced thymic activity [59], faster development of lymph node structures [60], accelerated maturation of memory T cells, and IgG class switched B cells [60]. Functional T cell responses against pp65 were generated in mice immunized with pp65 and multidimensional analyses of the T cell phenotypic markers by artificial neural network analyses demonstrated different immune signatures in the different lymphatic tissues analyzed [61]. This iDC cell therapy is currently being further modified to include additional antigens (gB, gH, gL) that can promote protective humoral responses against HCMV infection in humanized mice. Thus, the preclinical development of this immune cell therapy made specific use of humanized mice to demonstrate potency in this relevant model of human HCT before testing it in patients.

## 3. Preclinical Testing of Anti-HCMV Vaccines in HIS Mice

The serial challenges to generate potent protective vaccines against HCMV have been described during the last six decades (for a thorough overview of this topic, see [62]). So far, no vaccine has been approved, neither for HCT patients nor for newborns. A current strong candidate under clinical investigation is a modified vaccinia Ankara virus (MVA, Triplex) encoding pp65 and IE1. After a safety and feasibility phase-I clinical trial [36], this vaccine is currently being tested in phase-II clinical studies in the context of HCT (e.g., NCT03560752, NCT02506933). Meanwhile, the developers of this vaccine, Diamond et al. have worked on further improvements by including major humoral antigens, the glycoprotein B, and the gHgL-pentamer into the MVA [31]. Another mRNA-based vaccination for expression of gB plus the pentameric complex was also explored and the clinical trial is planned to start in 2020 (NCT04232280). Other promising vaccines, currently in preclinical development, are non-infective HCMV dense bodies carrying a variety of viral envelope antigens [33,63] and subunit vaccines based on the HCMV gHgLUL128-131 pentamer [32].

Humanized mice models of HCMV infection could be used to evaluate and compare these different vaccines in development before conducting extensive and expensive clinical studies. The ability to directly test the efficacy of vaccines by observing viral replication within these models might be very valuable to close the gap between preclinical and clinical development, which has occurred in the past and could be one reason why so many clinical studies did not reach their endpoints. Complementarily, it is a common ethical principle to validate new clinical developments in an appropriate model as exact as possible (in this case, with the human virus), before initiating trials in patients. However, a better understanding, improvement, and standardizations of these models among different laboratories are imperative.

In conclusion, optimizations of the currently available humanized mouse models are still needed in order to recapitulate all different stages of HCMV infection and the complex immunological host-pathogen interactions. This includes (i) improving the development and analyses of functional human immune responses; and (ii) taking in account the diverse latent and lytic human cell reservoirs that can be engrafted into the mice. Finally, academic collaborations towards standardization of the existing humanized mouse models can significantly accelerate this relevant preclinical translational field, towards the availability of robust and predictable models to efficiently test and compare different vaccines against HCMV.

## Figures and Tables

**Figure 1 vaccines-08-00089-f001:**
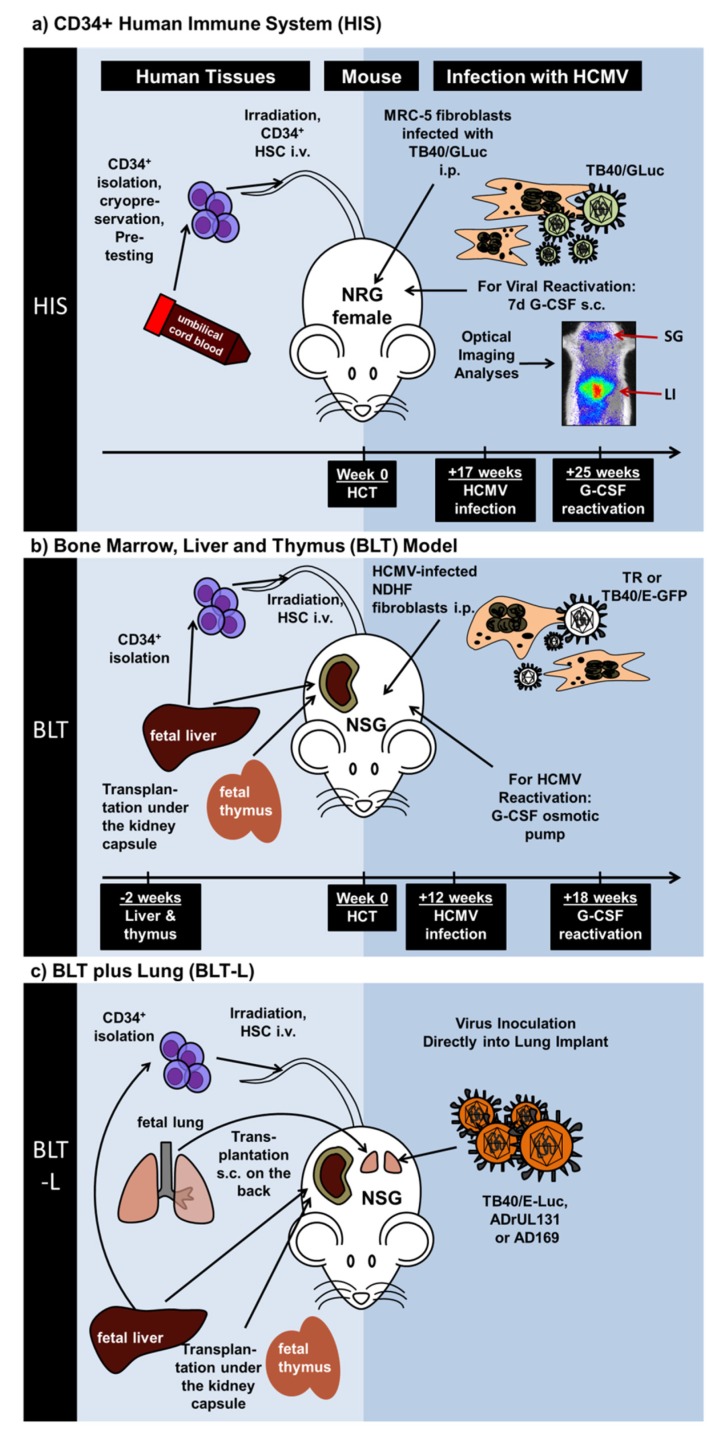
Schematic representation of recently described humanized mouse models exploring engraftment of different human tissues, immune-deficient mouse strains, and HCMV laboratory or clinical strains. (**a**) Mice with a human immune system (Theobald et al. (2018)). Cord blood-isolated CD34^+^ hematopoietic stem cells were transplanted i.v. into Nod.Rag.Gamma (NRG) mice. Seventeen weeks later, NRG mice with a developed human immune system (NRG-HIS) mice were administered i.p. with MRC-5 fibroblasts infected with the HCMV TB40 strain expressing Gaussia luciferase (GLuc). Eight weeks after infections, mice were administered daily with G-CSF s.c. for seven days to promote HCMV reactivation. Optical imaging analyses demonstrated the spread of the virus in different body parts such as liver (LI) and salivary glands (SG). (**b**) Bone marrow, liver and thymus (BLT) model (Crawford et al. (2017)). Human fetal tissues (liver and thymus) were used to generate the BLT model. Liver and fetal tissues were implanted under the kidney capsule of Nod.Scid.Gamma (NSG) mice. Two weeks later, CD34^+^ hematopoietic stem cells isolated from the liver were transplanted i.v. Twelve weeks after stem cell transplantation, NSG-BLT mice engrafted with the human tissues and developing human immune cells were administered with human non-dermal human fibroblasts (NDHF) infected with HCMV (strains TR or TB40-E-GFP). Six weeks after infections, an osmotic pump delivering G-CSF was used to promote HCMV reactivation. (**c**) BLT plus Lung (BLT-L) Model (Wahl et al. (2019)). Human fetal tissues (liver, thymus, lung) were implanted into Nod.Scid.Gamma (NSG) mice and CD34^+^ cells isolated from liver were used for stem cell transplantation. At various time-points after tissue engraftments, HCMV (strains TB40/ELuc, ADrUL131, or AD169) was inoculated directly into the engrafted lung.

**Table 1 vaccines-08-00089-t001:** Different modalities of in vivo models of CMV infection and their uses for vaccine testing purposes. Mouse CMV (MCMV), rhesus CMV (RhCMV), and human CMV (HCMV).

Model/Species	Virus Strain	Route of Infections	Immune Responses	Vaccine Evaluation
Immunocompetent mice	MCMV: e.g., Smith strain	intraplantar or	T, B, and NK cells responses against MV	Recombinant Adenovirus [21];
BALB/c; C57BL/6		Intraperitoneal (i.p.) injection		DNA vaccines [22,24];
(reviewed in [20])				Attenuated MCMV [23]
Rhesus macaques (reviewed in [25])	RhCMV	Subcutaneous (s.c.) or intravenous (i.v.) or vertical transmission	T and B cell responses, neutralizing antibodies against the Rh-Pentamer and gB	RhCMV-gB,
				-pp65 or -IE1 protein vaccines [36], modified vaccinia Ankara virus (MVA) expressing the RhCMV Pentamer [26]; RhCMV replication-defective virus [27], anti-RhCMV-IL10 vaccination [28]
Immunodeficient mice	HCMV TRpM1A [37] or	i.p. injection of in vitro infected fibroblasts	T cell responses against IE1, pp65, gB;	Not reported yet
NSG-HIS [37] or	TB40-GLuc [38]		IgGs responses against gB [38]	
NRG-HIS [38]				
NSG-huPBL CD34 [39]	HCMV from PBL donor	Infected CD34^+^ cells in PBL	Not reported	Not reported yet
NSG-A2-BLT [40]	HCMV Towne strain,	i.p. injection of in vitro infected fibroblasts [41,42] or DCs [40]	T cells against IE1, pp65 [40,42];	Live attenuated virus strain [40]
NSG-BLT [40,42,43]	TR-strain or		Specific IgM [40,42]	
	TB40E-GFP [41,42]		Specific IgG [42]	
	ULBP2-TB40 [40]			
NSG-BLT-L [44]	HCMV TB40/E-fLuc;	Lung implants were infected directly with virus injections	T cells against IE1 and pp65;	Not reported yet
	ADrUL131;			
	AD169		Specific IgM and IgG [44]	

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
