# Peer review of "Modeling Human Cytomegalovirus in Humanized Mice for Vaccine Testing"

_vaccines, 2020, doi:10.3390/vaccines8010089_

Round 1

Reviewer 1 Report

The manuscript "Modelling human cytomegalovirus in humanized mice for vaccine testing" by Koenig and colleagues describes the current progresses in the study of humanized mouse model of HCMV infections and the potential test of human vaccines before clinical use.

In my opinion, the present review is very well written, complete, provided by the appropriate references, clear and readable. The image and the table shown are very useful to further clarify and summarize the results obtained so far.

I have no objections. The manuscript deserves to be published in Vaccines.

Some minor comments: line 41 “cause mental retardation”, delete of line 66: “this is due to the fact”, add the line 162: “ NK cell numbers compared to mice”, add to line 247: “could be one reason why”, delete comma

Reviewer 2 Report

The manuscript entitled "Modelling human cytomegalovirus in humanized mice for vaccine testing" by Koenig et al provides an excellent description of the current status of the mice models available for CMV studies. The authors provide comprehensive experimental details as well as the strengthens and weaknesses of each model system. Given the current climate for the development of CMV vaccines, the paper is quite timely and would have the interest of the general audience. In general, it was well written and provided excellent insight into vaccine development using animal models.

Some minor comments that deal with syntax should be addressed on lines 23-25, 40-41, 66-67, 87-88, 127-128 (spell out 17), and 182-183 (Most complex is relative, please modify).
